# Glibenclamide does not improve outcome following severe collagenase-induced intracerebral hemorrhage in rats

Tiffany F. C. Kung[1], Cassandra M. Wilkinson[1], Christine A. Dirks[1], Glen C. Jickling[2,3], Frederick Colbourne[1,2]*

1 Department of Psychology, University of Alberta, Edmonton, Alberta, Canada, 2 Neuroscience and Mental Health Institute, University of Alberta, Edmonton, Alberta, Canada, 3 Division of Neurology, Faculty of Medicine, University of Alberta, Edmonton, Alberta, Canada

* fcolbour@ualberta.ca

**Data Availability Statement:** All relevant data are within the paper and its Supporting information files.

## Abstract

Intracerebral hemorrhage (ICH) is a devastating insult with few effective treatments. Edema and raised intracranial pressure contribute to poor outcome after ICH. Glibenclamide blocks the sulfonylurea 1 transient receptor potential melastatin 4 (Sur1-Trpm4) channel implicated in edema formation. While glibenclamide has been found to improve outcome and reduce mortality in animal models of severe ischemic stroke, in ICH the effects are less clear. In our previous study, we found no benefit after a moderate-sized bleed, while others have reported benefit. Here we tested the hypothesis that glibenclamide may only be effective in severe ICH, where edema is an important contributor to outcome. Glibenclamide (10 μg/kg loading dose, 200 ng/h continuous infusion) was administered 2 hours post-ICH induced by collagenase injection into the striatum of adult rats. A survival period of 24 hours was maintained for experiments 1–3, and 72 hours for experiment 4. Glibenclamide did not affect hematoma volume (~81 μL) or other safety endpoints (e.g., glucose levels), suggesting the drug is safe. However, glibenclamide did not lessen striatal edema (~83% brain water content), ionic dyshomeostasis (Na⁺, K⁺), or functional impairment (e.g., neurological deficits (median = 10 out of 14), etc.) at 24 hours. It also did not affect edema at 72 h (~86% brain water content), or overall mortality rates (25% and 29.4% overall in vehicle vs. glibenclamide-treated severe strokes). Furthermore, glibenclamide appears to worsen cytotoxic edema in the peri-hematoma region (cell bodies were 46% larger at 24 h, p = 0.0017), but no effect on cell volume or density was noted elsewhere. Overall, these findings refute our hypothesis, as glibenclamide produced no favorable effects following severe ICH.

## Introduction

Intracerebral hemorrhage (ICH) is a devastating stroke subtype with a mortality rate of ~40%, for which no neuroprotective treatments have been found [1–4]. In addition to the immediate primary damage caused by the bleed, secondary damage mechanisms continue to exacerbate

**Funding:** This research was funded by a Canadian Institutes of Health Research (CIHR) project grant to F. C. C.W. is supported by a CIHR doctoral award, an Isaak Walton Killam Memorial scholarship, and a Dorothy J. Killam Memorial Scholarship. T.K. is supported by an Alberta Innovates (AI) Summer Research Studentship and a CIHR CGS-M. C.D. is supported by an Undergraduate Research Initiative (URI) stipend, and F.C. is a Canada Research Chair in ICH. The funders had no role in study design, data collection and analysis, decision to publish, or preparation of the manuscript.

**Competing interests:** The authors have declared that no competing interests exist.

injury for weeks following the initial stroke [1, 2]. Reducing secondary damage (e.g., edema) is a potential target to improve outcome and reduce mortality. Many focus on cerebral edema, which emerges shortly after stroke onset as excess water accumulates in the brain parenchyma. In the collagenase-induced ICH rat model, edema peaks in 1–3 days, and resolves within 7 days [5–9]. Of the various forms of edema, cytotoxic edema is often the earliest, wherein cells increase ion uptake. This is then accompanied by osmotic influx of water into cells, and various ion and water channels have been implicated in this process (e.g., aquaporin-4, Na-K-Cl co-transporter, and sulfonylurea 1 transient receptor potential melastatin 4 (Sur1-Trpm4)) [2, 10, 11]. Cytotoxic edema is driven by ionic dyshomeostasis, and worsens injury by further aggravating ionic dyshomeostasis, cell swelling, and blood-brain barrier (BBB) damage [2, 10–13]. This then culminates in worsened vasogenic edema [2, 10]. The added mass caused by edema can then further raise intracranial pressure (ICP), leading to midline shift, brain herniation, and mortality [11, 14]. Thus, edema-reducing treatments should additionally confer benefit by reducing ICP.

The Monro-Kellie doctrine describes ICP regulation and dictates that normally, the brain, blood, and cerebrospinal fluid (CSF) must remain in equilibrium [15]. Following ICH, cerebral venous blood and CSF volumes are redirected out of the skull to compensate for the added mass of the hematoma and to offset rising ICP (i.e., volume buffering) [16]. In addition, we have evidence for tissue compliance wherein neurons become smaller (i.e., decreased volume) and increase their packing density throughout the brain to compensate for mass effects and to minimize ICP elevations, at least after severe strokes [14, 17]. Undoubtedly, these cell volume changes are regulated by ion channels, and therefore existing therapies likely impact not only peri-hematoma edema, but also the volume of the remaining brain [12, 18, 19].

In ischemic stroke, glibenclamide reduces mortality and improves outcome in rodent pre-clinical studies [20, 21]. In humans, glibenclamide is currently used to treat Type II diabetes. For human ischemic stroke patients, a Phase II trial of glibenclamide found no benefit on the primary endpoint (modified Rankin scale for neurological disability score at 90 days), but a sub-group analysis suggested some benefit to patients with large hemispheric strokes at a high risk of edema [22]. Glibenclamide is currently in a Phase III trial as a combined therapy with tissue plasminogen activator (rt-PA) in cerebral embolism [23]. Glibenclamide works by blocking Sur1-Trpm4, a transient, non-selective monovalent cation channel that is not constitutively expressed, but upregulated after stroke [24]. Sur1-Trpm4 is thought to contribute to cytotoxic edema by facilitating the influx of $Na^+$ and efflux of $K^+$ from cells [11, 25]. Though many negatively-charged macromolecules bind to $K^+$ within the cell, this is not the case for $Na^+$, meaning $Na^+$ influx (compared to $K^+$ efflux) disproportionately generates a strong osmotic gradient for water influx [10, 11]. Trp non-selective cation channels, and specifically Sur1-Trpm4, are thought to be involved in cell volume regulation after ischemic stroke [11, 12, 25]. In pre-clinical studies, blocking the Sur1-Trpm4 channel with glibenclamide attenuates swelling following severe ischemic stroke, demonstrating the significance of the channel in edema formation [25].

Given the promising results in ischemic stroke, several groups have evaluated whether glibenclamide reduces edema and improves recovery and survival after ICH. Notably, Jiang et al. found that glibenclamide improved performance on both the Morris Water Maze and the modified neurological severity score (mNSS), protected BBB integrity, and lessened edema in an autologous whole blood model of rat ICH [26]. In the same model, Xu et al. found glibenclamide inhibited the NLR family pyrin domain containing 3 (NLRP3) inflammasome, which they hypothesized subsequently improved BBB integrity, reduced edema, and improved Garcia score and rotarod test performance in mice [27]. Zhou et al. found that glibenclamide reduced oxidative stress and improved mNSS in a collagenase-induced rat ICH model with

~25 μL hematoma volume at 72 hours [28]. Conversely, Wilkinson et al. reported that gliben-clamide did not improve scores on the Montoya staircase, horizontal-ladder walking test, or NDS, and did not improve edema, BBB integrity, or ion dyshomeostasis in a collagenase model of mild (~15 μL hematoma volume) rat ICH [29].

There are a number of potential reasons for the discrepancies among studies, specifically why Wilkinson et al. failed to observe benefit [29]. It may be that glibenclamide is only effec-tive when administered early, or prior to ICH, as done by both Xu et al. and Zhou et al., which limits clinical relevance [27, 28]. However, Wilkinson et al. had the same dosing regimen as Jiang et al., who found benefit. Additionally, Jiang et al., Xu et al., and Wilkinson et al. all assessed edema at 72 hours [26, 27, 29]. One possibility for this difference is insult severity, which would be considered mild in Wilkinson et al.'s study (~15 μL hematoma volume) [29]. Specifically, the benefits of glibenclamide may be more apparent following severe stroke where edema is greater, and thus reducing edema will have a larger impact on outcome and mortality.

In this study, we assess whether glibenclamide improves outcome after moderate (~40 μL) and severe (~80 μL) collagenase-induced ICH in rat. The collagenase model was chosen to maintain consistency with our previous study [29] and because the model produces more edema than the autologous whole blood model [30–32]. In our first experiment, we hypothe-sized glibenclamide would not affect cerebral bleeding caused by collagenase, or other safety measures (e.g., blood glucose) at 24 hours [29, 33]. In experiment 2, we hypothesized glibencla-mide would reduce edema and improve related factors, including BBB integrity and element dyshomeostasis at 24 hours [2, 10, 11, 13]. Two different stroke severities were administered, deemed to be moderate and severe, to further investigate the effect of glibenclamide in relation to insult severity. In experiment 3, we hypothesized glibenclamide would reduce injury volume and indirectly decrease cell shrinkage (compression of distal structures) at 24 hours by lower-ing edema and reducing the pressure exerted on remote structures. Lastly, in experiment 4, we hypothesized glibenclamide would reduce edema and improve functional outcome at 72 hours. Throughout all experiments, animals were assessed for functional outcome using the neurological deficit scale (NDS) and forelimb placing tests, common measures of functional recovery in stroke models [34, 35].

## Materials and methods

### Ethics

All procedures were conducted in accordance with the Canadian Council on Animal Care guidelines and approved by the Biosciences Animal Care and Use Committee at the University of Alberta (protocol: AUP960). To reduce suffering, end points for experiments 1–3 were kept to a survival time of 24 hours and animals were monitored repeatedly. For experiment 4, guidelines for premature euthanasia were established and monitoring was done repeatedly over the 3-day period.

### Subjects

One-hundred and six male Sprague Dawley rats (~300–400 g, ~3–4 months old) from Charles River Laboratory (Saint Constant, Quebec) were used for this project. Rats were housed 3 to 4 per cage in a temperature- and light-controlled environment (12 h light cycle). Rats were given free access to food and water and used during the lights-on phase of their cycle. Following ICH, animals were housed individually. Treatment (glibenclamide or vehicle) and severity (severe or moderate) were randomized using random.org, and for all experiments, another

researcher prepared the drug or vehicle for each animal to ensure blinding. All animals were kept in the same location and were not housed by group.

## Experimental design

At least 24 hours prior to behavioural baseline (-3d), all animals were handled for two ten-minute periods to gain familiarity with the researcher. All rats additionally received one NDS training session and two forelimb placing training sessions to gain familiarity with the specific behavioural tests. Behavioural baseline was taken two days prior to ICH induction in the left striatum (-2d). Functional outcome was assessed at 24 hours (experiments 1–3) or 72 hours (experiment 4) via NDS and the forelimb placing test as recommended and previously done [29, 34, 35]. Rats were euthanized shortly after, at 24 hours (experiments 1–3) or 72 hours (experiment 4) post-ICH.

In an effort to ensure translational rigour, this study adhered to an *a priori* planning document, except experiment 4, which was conducted as a post-hoc follow up to the 24-hour experiments. Statistical power and tests were planned beforehand, and any deviations from that plan are explicitly noted (none impacted our conclusions).

**Experiment 1.** Hematoma volume was our *a priori* primary endpoint for this experiment and was assessed with a hemoglobin assay at 24 hours (Fig 1A). Secondary endpoints, including blood glucose, behaviour, and rat grimace scale (RGS), were assessed at 23 hours. Behavioural outcomes were assessed at baseline (2 days prior to ICH, -2d) and just prior to

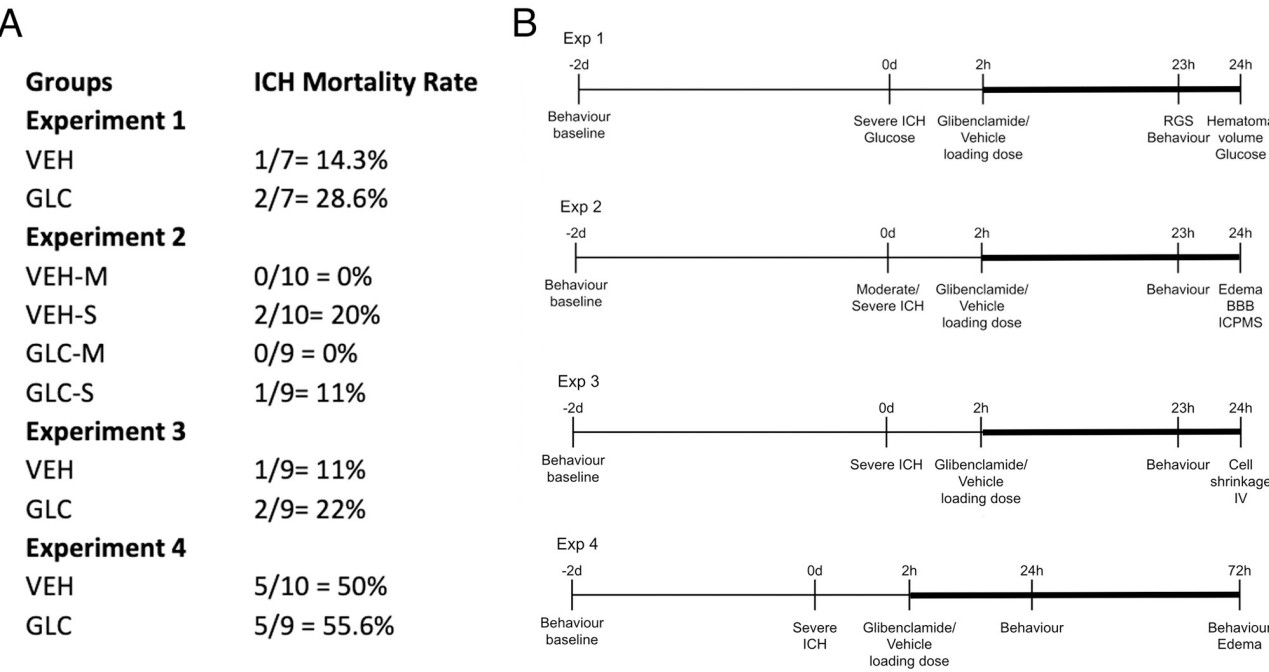

**Fig 1. Mortalities and timelines.** Mortality rates per experiment for vehicle-treated (VEH) and glibenclamide-treated (GLC) groups (A). Experimental timelines are also shown (B). The continuous glibenclamide infusion period is denoted by thick bars. Mortality rates for severe strokes were not significantly different across groups (p = 0.6783). There were no mortalities for the moderate stroke group. Exclusions for each experiment are listed in the Results section. RGS, hematoma volume and blood glucose were assessed in experiment 1. Edema, BBB integrity, and element concentrations (via inductively coupled mass spectrometry (ICPMS)) were assessed in experiment 2. Cell volume and density along with injury volume assessments were performed in experiment 3. Edema was assessed in experiment 4. Behaviour, as assessed by the NDS and forelimb placing, was performed for all experiments at 24 hours (experiments 1–3) and 72 hours (experiment 4). All rats had a behavioural baseline taken 2 days prior to ICH, and drug infusion began 2 hours post-ICH induction. Rats were euthanized at 24 hours (experiments 1–3) or 72 hours (experiment 4).

euthanasia (24 h). Food and water intake were assessed repeatedly from behavioural baseline (-2d) to euthanasia (24 h). All rats were given severe strokes and randomly assigned to either glibenclamide (GLC) or vehicle (VEH) (n = 7 each).

**Experiment 2.** Edema was our *a priori* primary endpoint and was assessed via the wet-weight dry-weight method at 24 hours (Fig 1A). Element concentrations and BBB integrity were assessed at 24 hours. Behaviour was assessed as described above. Rats were given either vehicle (VEH) or glibenclamide (GLC) after either a moderate (-M) or severe (-S) ICH, with n = 10 per group.

**Experiment 3.** This experiment used severe strokes, as glibenclamide had a trend toward greater benefit on this group (as per our *a priori* research plan). The *a priori* primary endpoint for the experiment was cell volume and density measurements at 24 hours (Fig 1A). Injury volume was also analyzed to quantify stroke severity. Behaviour was assessed as described above, and the experiment had n = 9 per group.

**Experiment 4.** Edema was our primary endpoint for this experiment and was assessed via the wet-weight dry-weight method at 72 hours (Fig 1A). Behaviour was assessed as described above at both 24 and 72 h, and the experiment had n = 12 per group.

**Additional data.** Microarray data from a previous study (unpublished) were analyzed and used to investigate the expression of Sur1-Trpm4 post-ICH. In that study, 10 animals (n = 5 per group) were randomly assigned to sham or collagenase ICH similar to our moderate stroke (~40 μL). At 24 hours after ICH, animals were euthanized and ipsilateral cortex (adjacent to the hematoma, peri-hematoma zone) as well as contralateral CA1 (a region previously shown to undergo cell compression after moderate to large bleeds) were dissected and flash frozen. RNA was extracted using a trizol-based extraction kit (Direct-zol, Zymo Research). Samples were analyzed with Clariom S microarray (Thermo Fisher Scientific) and data was analyzed using Partek (v7.0, St. Louis, MO). Here, we present the data for expression of *Abcc8* (codes for Sur1) and *Trpm4* in both regions.

## Stroke induction

Bacterial collagenase (Type IV-S, Sigma, 0.2 U/μL in saline, 2.25 μL for moderate stroke and 4 μL for severe stroke), was injected into the left striatum at 0.5 mm anterior, 3.5 mm lateral (left), and 6.5 mm below Bregma, as described previously. Rats were anesthetized with isoflurane (4% induction, 2% maintenance in 70% $N_2O$ and balance $O_2$) and bupivacaine hydrochloride (Steri*Max* Inc., Oakville, ON) was delivered subcutaneously prior to the initial incision, and after ICH induction (0.1 mL for both). Rectal probes were inserted to monitor and maintain body temperature at ~37˚C throughout surgery. Immediately following surgery, rats were given 5 mL S.C. of saline and wet mashed up Purina rat chow to reduce risk of dehydration. Subsequent injections of saline, as well as cake batter (Betty Crocker™, Chocolate Fudge flavour) mixed with water were given in rats that displayed signs of continued dehydration or reduced feeding.

## Glibenclamide administration

Glibenclamide was prepared and administered as described previously (loading dose 10 μg/kg, 200 ng/h continuous infusion) [29]. A loading dose of glibenclamide was given intraperitoneally 2 hours after collagenase infusion, and mini osmotic pumps (Alzet osmotic pumps, model 2001, 1.0 μL/h) were surgically implanted in the nape of the neck for continuous infusion at the same time, a dosing regimen that is commonly used [26, 29, 36, 37]. Drug delivery was confirmed by measuring the amount of solution left in the extracted mini osmotic pumps. In all cases, the drug infused properly.

## Health measures

Food and water intake were assessed by subtracting post-ICH intake from baseline averages (-1d and -2d). Blood glucose was measured immediately prior to ICH, and again just before euthanasia. For RGS, a 10-minute video was recorded, and snapshots were taken each minute when the rat's face was clearly visible [29, 33]. Orbital tightening, nose/cheek flattening, ear changes, and whisker changes were each assessed as indicators of pain on a scale from 0 (no pain) to 2. Scores were then summated per category for a total score from 0 to 10, then further averaged per animal [33].

## Functional/Behavioural outcome

For all behavioural assessments, a baseline measurement was taken two days before ICH induction (day -2), and assessments were conducted at 23–24 hours post-ICH (all experiments) and 72 hours (experiment 4) after ICH. Spontaneous circling, contralateral hind limb retraction, bilateral forepaw grasp, beam walking ability, and contralateral forelimb flexion were assessed in NDS, and rats received composite scores ranging from 0 (no deficit) to 14 (severely impaired) as previously described [29]. Vibrissae-evoked forelimb placing was assessed for the left paw and right paw by stimulating the ipsilateral whiskers (e.g., left whisker stroked, left paw placement). Midline left paw and midline right paw was assessed by stimulating the contralateral whiskers and restraining the ipsilateral paw (e.g., left whisker stroked, left paw restrained, right paw placement).

## Hemoglobin assay

Hemoglobin assays were done as previously described with modification [29, 38]. The ipsilateral and contralateral hemispheres were separated and diluted 1:4 in distilled water, homogenized and centrifuged. The supernatant was combined 1:6 with Drabkin's reagent, given 15 minutes to react, and blood volume was then measured using spectrophotometry at 540 nm.

## Edema

Edema was measured using the wet-weight dry-weight method, similar to our previous study [29]. Brains were blocked from 2 mm anterior of the collagenase injection site to 4 mm posterior of the collagenase injection site. Fresh brains were extracted and split into ipsilateral and contralateral striatum and cortex, and cerebellum. They were then weighed (wet weight), baked for 24 h at 100˚C and re-weighed (dry weight).

## Blood brain barrier integrity and ICPMS

Clinical-grade gadopentetate dimeglumine (Gd) was injected into the rats' tail vein, and allowed to circulate for 10 minutes prior to euthanasia by decapitation under isoflurane anesthesia [29]. For all groups, 5 brains were sent to the Canadian Center for Isotopic Microanalysis to be analyzed for gadopentate concentration, as well as element concentrations (Na, K, Fe).

## Histology

After administering ~100 mg/kg i.p. of pentobarbital (Bimeda MTC, Cambridge ON) at 24 hours, the animals were perfused with saline followed by 10% neutral buffered formalin (Fisher) and brains were extracted. At least 48 hours prior to sectioning, brains were transferred to 30% sucrose in formalin, and later sectioned on a cryostat (20 μm). ImageJ was used to quantify cell density, cell volume, and injury volume. For cell density, average neuron

counts were divided by 3.78 to determine the average number of cells per brain volume fraction of $1.0 \times 10^{-4}$ $mm^3$ as previously described [17]. For cell volume, five cells were chosen by a random offset (measurement) grid, and were assessed if the nucleolus was aligned with a point intersection [17, 39–44]. Maximum injury volume was assessed by measuring 4 mm around (anterior and posterior) the maximum injury area. Rats were included in the analysis as long as they had a total of 4 mm around the maximum volume, that was split with at least 0.8 mm measurable on any one given side.

## Statistical analysis

Data were analyzed using GraphPad Prism (v 8.4.0 for MacOS, GraphPad Software Inc., San Diego, California USA). All data are presented as either mean ± 95% confidence intervals (CI), or median ± inter-quartile ranges (IQR), as indicated. Statistical analyses were conducted as per our *a priori* planning document, except where explicitly indicated below, to improve statistical accuracy. Sample sizes were calculated to give 80% power to detect a moderate to large effect size (30% change in hematoma volume, 1.5% absolute decrease in edema, 25% change in cell shrinkage, based upon variability measurements from our recent work in this area) [14]. For experiment 4, power was calculated as done for experiment 2, but with increased group sizes due to greater anticipated mortality. Hematoma volume, blood glucose, food and water intake (experiment 1), as well as cell volume and density for CA1 and CA3 and injury volume (experiment 3) were analyzed using an independent-samples two-tailed t-test. CA1 and CA3 cell shrinkage was not analyzed with a two-way ANOVA as planned, because the ipsilateral hemisphere was not measurable. RGS (experiment 1), NDS and forelimb placing (experiments 1, 3, and pooled analysis) were analyzed using a Mann-Whitney test. NDS and forelimb placing (experiment 2) were analyzed using Kruskal-Wallis. Contrary to planned in our *a priori* document, RGS was analyzed with Mann-Whitney U test rather than an unpaired t-test as the first is a more appropriate test for ordinal data. Edema, ICPMS, BBB integrity (experiment 2) and cell shrinkage for S1 (experiment 3) were analyzed using a two-way ANOVA with Fisher's LSD for more sensitivity to detect benefit (minimize Type II errors). An unplanned analysis of average pooled behaviour ranks was done as previously described [45], and analyzed using a Mann-Whitney. Body weight was pooled across experiments and analyzed using a two-way ANOVA. For experiment 4, NDS was analyzed with a Mann-Whitney at each time point, and forelimb placing for each paw was analyzed using Kruskal-Wallis. Edema was analyzed using a 2-way ANOVA with Fisher's LSD for more sensitivity to detect benefit. Mortality was pooled across experiments and analyzed using a chi-squared test. Channel expression was analyzed using a two-way ANOVA with group and region as factors. Cohen's d was used to calculate effect size between groups, and Cohen's $d_z$ for within groups, for all significant interactions using G*Power (version 3.1).

## Results

### Experiment 1

**Mortality and exclusions.** A total of 14 animals were used for experiment 1 (n = 7 per group). Of these, 3 animals spontaneously died (2 GLC, 1 VEH), 1 of which was excluded from hematoma volume analysis (GLC). The other 2 rats (1 GLC, 1 VEH) had their brains processed as soon as possible post-mortem. All 3 animals were excluded from secondary endpoints due to a lack of behavioural and safety data at 24 hours post-ICH. For blood glucose, 1 animal (VEH) was excluded due to an extended period of isoflurane administration, which is known to increase blood glucose and thus was a potential confound [46]. For food intake, 1 animal (GLC) was excluded due to support staff error. For forelimb analysis, six measurements

were excluded due to poor baseline performance, possibly due to insufficient handling prior to testing, which impacted testing scores, determined as a score of <8 at baseline (1 VEH left and right, 3 VEH left and midline right, 1 GLC midline left).

**Behaviour.**  ICH significantly worsened NDS scores (p<0.0001 vs. baseline, Cohen's $d_z$ = 6.48), but was not affected by glibenclamide (Fig 2A, p = 0.4091). ICH also significantly worsened forelimb placing for all limbs (Fig 2B, each paw p<0.0001, Cohen's $d_z$ = 1.59), and was not affected by glibenclamide (p≥0.1190).

**Hematoma volume.**  Hematoma volume (blood volume in the ipsilateral—contralateral hemisphere) averaged 82.6 μL and 79.1 μL in the VEH and GLC groups respectively, which was not significantly different (Fig 2C, p = 0.8333).

**Safety measures.**  Blood glucose was lower at 24 hours after ICH than baseline (p = 0.0139, Cohen's $d_z$ = 1.51), likely due to decreased food intake, but values were still in the normal range. Glibenclamide did not affect blood glucose 24 hours post-ICH (Fig 3A, p = 0.2518). No difference was found in RGS between groups 23 hours after ICH (Fig 3B, p = 0.4286), indicating that treatment did not cause or mitigate pain responses. Food and water intake were reduced at 24 hours compared to baseline (p = 0.0079 and p<0.0001, respectively), but there were no group differences (Fig 3C and 3D; p = 0.7654 and p = 0.3763, respectively). All animals fed mostly from the moist food mash provided in their cage. Thus, we likely underestimated water intake as that was measured solely from the water bottle hanging atop their cage.

However, glibenclamide did not affect blood glucose (p = 0.2518), RGS (p = 0.4286), food intake (p = 0.7654), or water intake (p = 0.3763).

## Experiment 2

**Mortality and exclusions.**  A total of 40 animals were used in experiment 2 (n = 10 per group). Of these, 3 animals spontaneously died (2 VEH-S, 1 GLC-S) and were excluded from all analyses except mortality. For edema analyses, 4 animals were excluded, 1 due to hematoma leaking out during brain extraction (GLC-M), 1 due to technical error (VEH-S), and 2 who experienced intra-ventricular hemorrhage rather than ICH (GLC-S, GLC-M). One VEH-M and one GLC-S rat were excluded from NDS due to technical issues. The ipsilateral and contralateral striatum of 5 rats (20 animals, 40 samples total) were sent for ICPMS analysis, 1 of which (GLC-M) was excluded for experiencing intra-ventricular hemorrhage rather than ICH (same animal as above).

**Behaviour.**  ICH significantly impaired NDS (p<0.0001 vs. baseline, Cohen's $d_z$ = 9.72) and forelimb placing (p<0.0001 vs. baseline, Cohen's $d_z$ = 1.47). Glibenclamide had no significant effect on NDS (Fig 4A, p = 0.1133) or forelimb placing of any paw (Fig 4B, p≥0.1131). There was a trend towards better NDS scores for the VEH-M group when compared to GLIB-M (p = 0.0558). In order to detect this effect with 80% power, group sizes of 19 would have been required. We calculated our post-hoc power for this non-significant effect as 47.1%. This effect was only observed between VEH-M and GLIB-M, and it is unclear whether an effect of this size is biologically meaningful.

**Edema.**  ICH significantly increased striatal brain water content to ~83% (vs. 79% contralaterally, p<0.0001, Cohen's $d_z$ = 2.94) for all groups, but was not affected by glibenclamide (Fig 4C, p = 0.8040). There was some evidence (trends) for cortical edema, but only the GLC-S group had significantly higher brain water content (vs. contralateral cortex, p = 0.0191). Glibenclamide did not significantly affect brain water content in ipsilateral cortex (p = 0.5143).

**ICPMS.**  There was no difference in gadopentetate dimeglumine (Gd, measure of BBB integrity) between groups (p = 0.8061), though a trend was observed between hemispheres

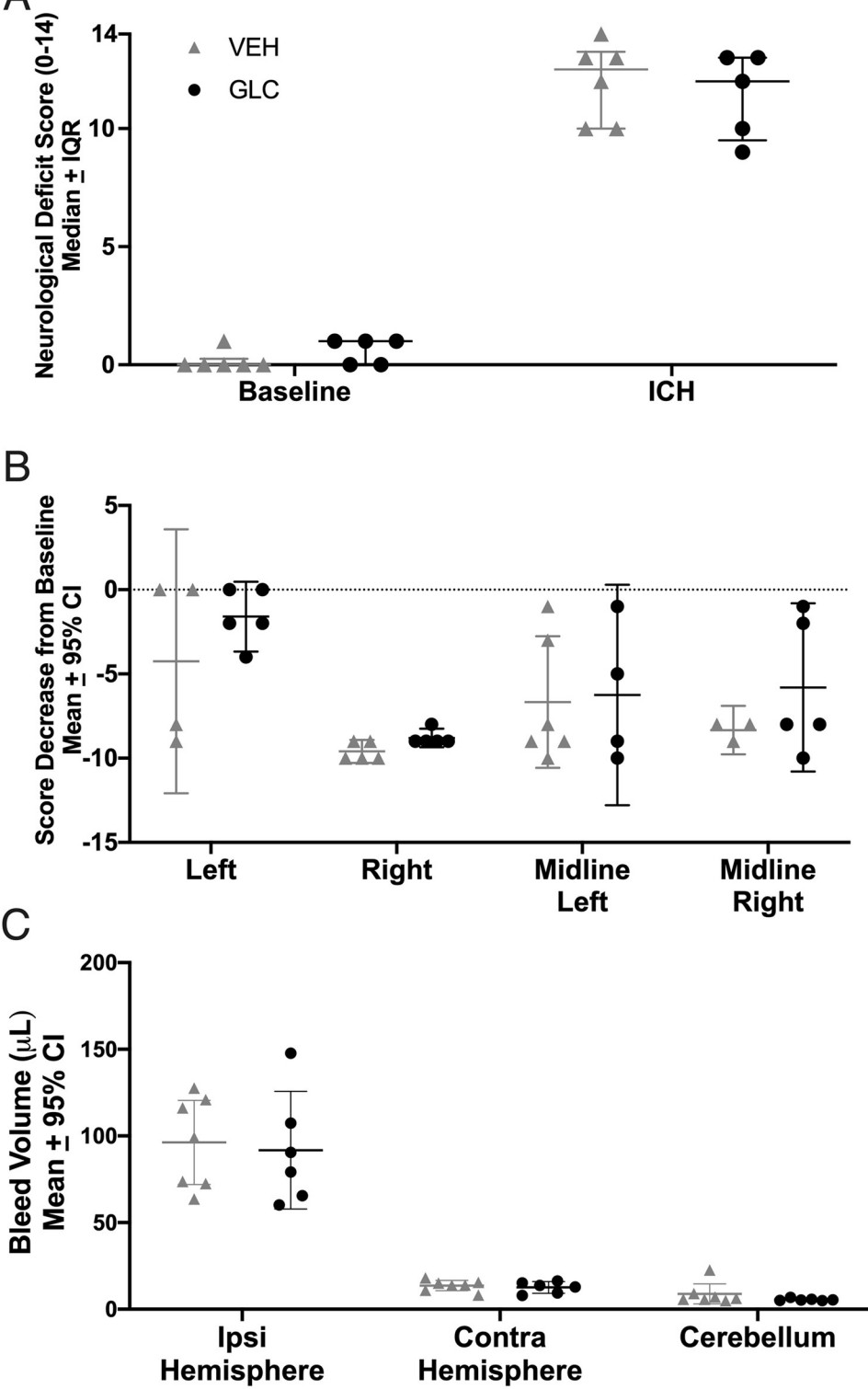

**Fig 2. Experiment 1 behaviour and hematoma volume.** ICH significantly worsened NDS (A, p<0.0001) and forelimb placing (B, p<0.0001) scores for all animals. No benefits of glibenclamide were found in NDS (p = 0.4091), forelimb placing for any paw (p≥0.1190), or hematoma volume (ipsilateral—contralateral hemisphere; C, p = 0.8333). For forelimb placing, a dashed line was set at 0, which indicates no change from baseline. Collagenase caused a significant bleed amounting to 96.2 µL and 91.7 µL in the ipsilateral striatum (not adjusted to contralateral hemisphere) of VEH and GLC animals, respectively.

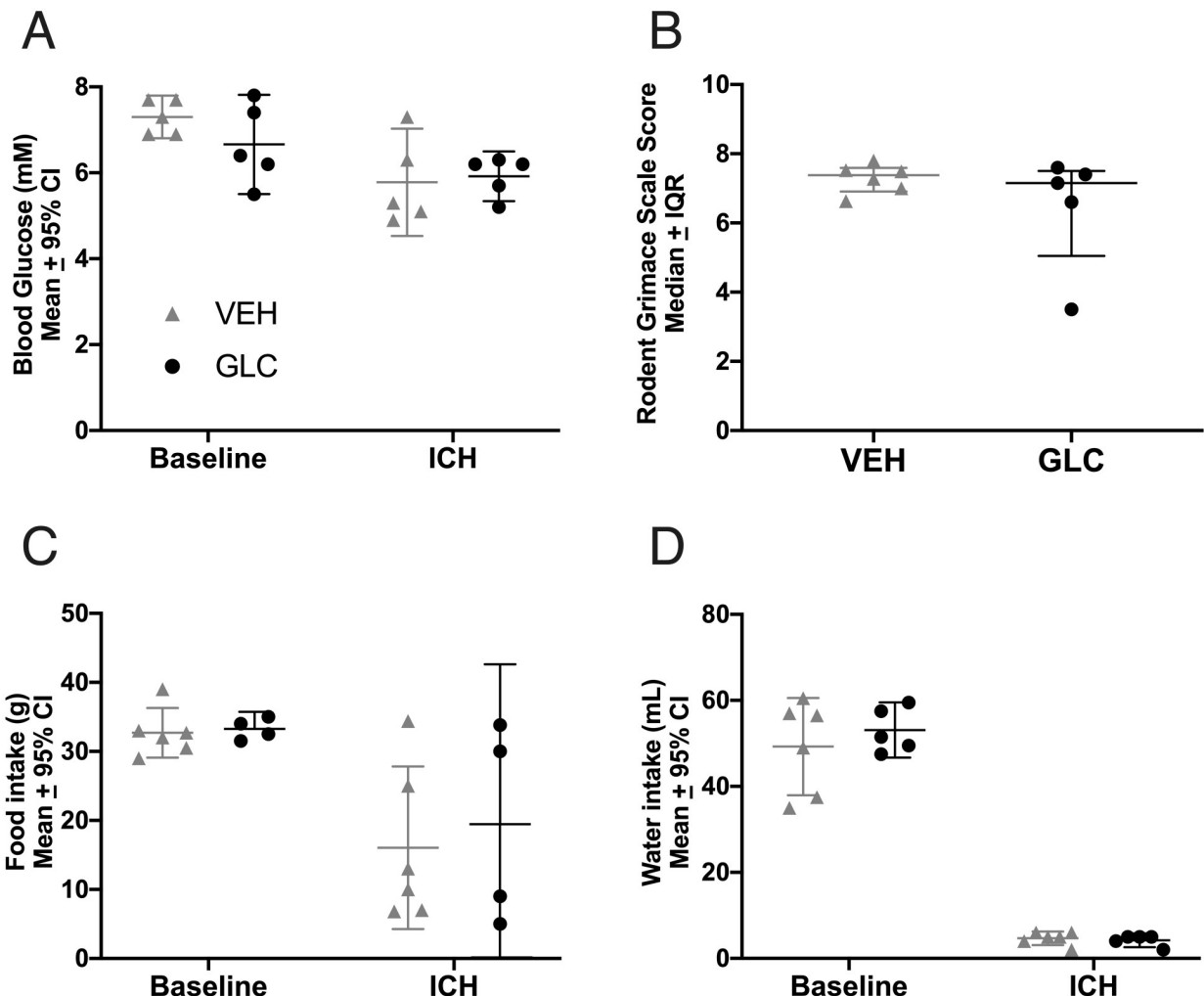

**Fig 3. Experiment 1 safety measures.** Blood glucose (A, p = 0.0139), RGS (B, p = 0.0139), food intake (C, p = 0.0079), and water intake (D, p<0.0001) were all decreased following ICH.

(Fig 5A, p = 0.0611). Fe (an estimate of hematoma volume) was significantly higher in the ipsilateral striatum (p<0.0001 vs. contralateral striatum, Cohen's $d_z$ = 7.49) but not between groups (Fig 5B, p = 0.9466). $Na^+$ (p = 0.0033, Cohen's $d_z$ = 2.31) and $K^+$ (p<0.0001, Cohen's $d_z$ = 2.87) were significantly different between hemispheres but not between groups (Fig 5C and 5D, p≥0.6542).

### Experiment 3

**Mortality and exclusions.** A total of 18 animals were used for experiment 3 (n = 9 per group). Of these, 3 animals spontaneously died (2 GLC, 1 VEH) and were excluded from secondary endpoints. In all rats, damage frequently extended into ipsilateral hippocampus, so only contralateral CA1 and CA3 cell volume and density were assessed. An additional 4 animals (3 VEH, 1 GLC) were excluded from CA1 analysis, and 3 animals (VEH) were excluded from CA3 analyses based upon *a priori* criteria (e.g., histological artefacts). Both ipsilateral and contralateral S1 were assessed. For injury volume analysis, 1 additional animal was excluded for not meeting the requirements listed in the methods.

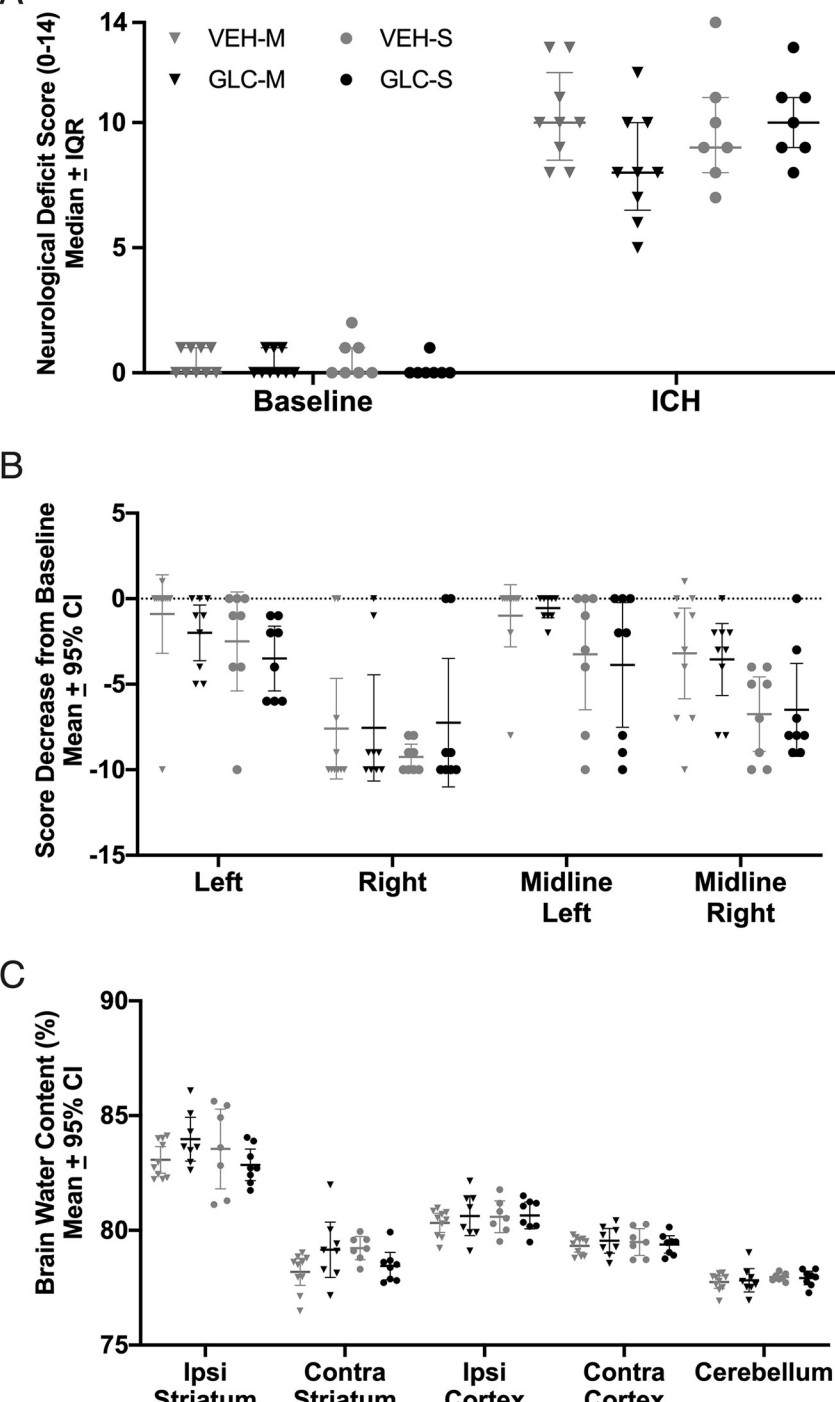

**Fig 4. Experiment 2 behaviour and edema.** All groups had increased brain water content in the ipsilateral striatum compared to the contralateral striatum (p<0.0001). No meaningful benefit of glibenclamide was found for NDS (A, p = 0.1133) or forelimb placing of any paw (B, p≥0.1131) in either moderate (-M) or severe (-S) bleeds. Although VEH-M rats had somewhat better NDS scores, the differences were not significant (C, p = 0.0558). Glibenclamide also did not have an effect on brain water content in either severity (p = 0.8040). For forelimb placing, a dashed line was set at 0, which indicates no change from baseline.

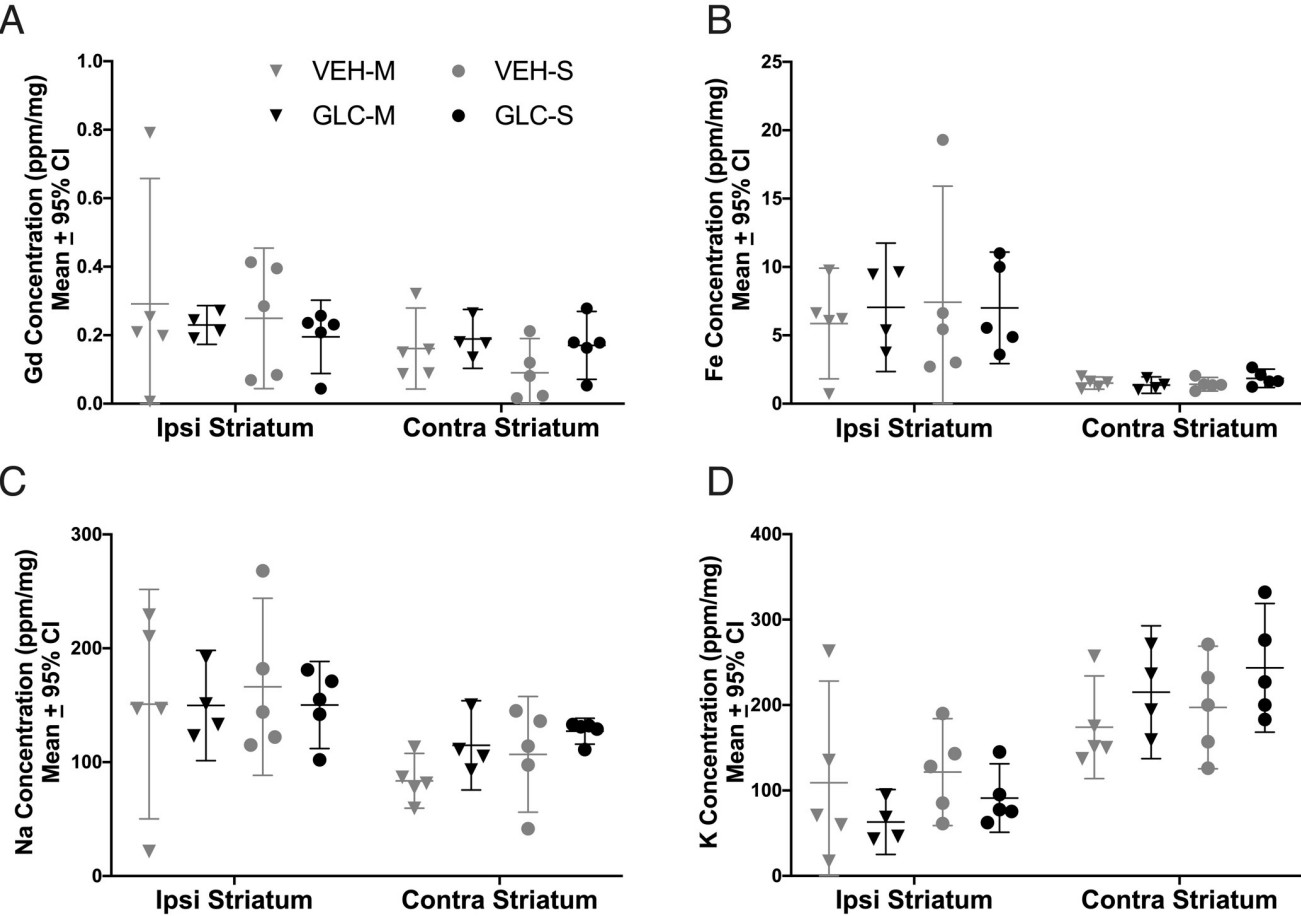

**Fig 5. Experiment 2 ICPMS and BBB integrity.** Gd (A), Fe (B), Na (C) and K (D) concentrations were analyzed using ICPMS. After ICH, Gd was not significantly different between hemispheres (A, p = 0.0611). Fe (B, p<0.0001) and Na (C, p = 0.0033) were significantly increased in the ipsilateral hemisphere, and K (D, p<0.0001) was significantly decreased in the ipsilateral hemisphere. Glibenclamide did not affect Gd (p = 0.8061), a measure of BBB integrity, Fe (p = 0.9466), an estimator for hematoma volume, Na (p = 0.7027), or K (p = 0.6542).

**Behaviour.** ICH impaired performance on NDS (p<0.0001 vs. baseline, Cohen's $d_z$ = 16.77) and forelimb placing tasks (p<0.0001 vs. baseline, Cohen's $d_z$ = 1.83). Glibenclamide did not affect performance on NDS (Fig 6A, p = 0.5970) or forelimb placing (Fig 6B, p≥0.4432).

**Injury volume assessment.** Extensive damage was seen in the basal ganglia, and some other structures were destroyed. No significant difference was found between groups for injury volume (Fig 6C, p = 0.1280). Representative sections are shown (Fig 6D).

**Cell volume and density.** In the peri-hematoma S1 region, there was a significant group effect on cell density (p = 0.0132), where glibenclamide treated rats had lower cell density in ipsilateral S1 (Fig 7A, p = 0.037, Cohen's d = 1.19) but not in the contralateral S1 (p = 0.1343). Glibenclamide-treated rats also had higher cell volume in the ipsilateral hemisphere (Fig 7B, p = 0.0017, Cohen's d = 2.18). but not the contralateral hemisphere (p = 0.7052). Both contra-lateral CA3 and CA1 cell density and volume were not significantly affected by glibenclamide administration (Fig 7C–7F, p≥0.2276).

## Pooled analysis for 24 h survival time data

A pooled analysis was done to improve sensitivity for smaller potential effects of glibenclamide treatment. All severe stroke rats (experiments 1–3) were pooled for analysis (n = 26 GLC-S,

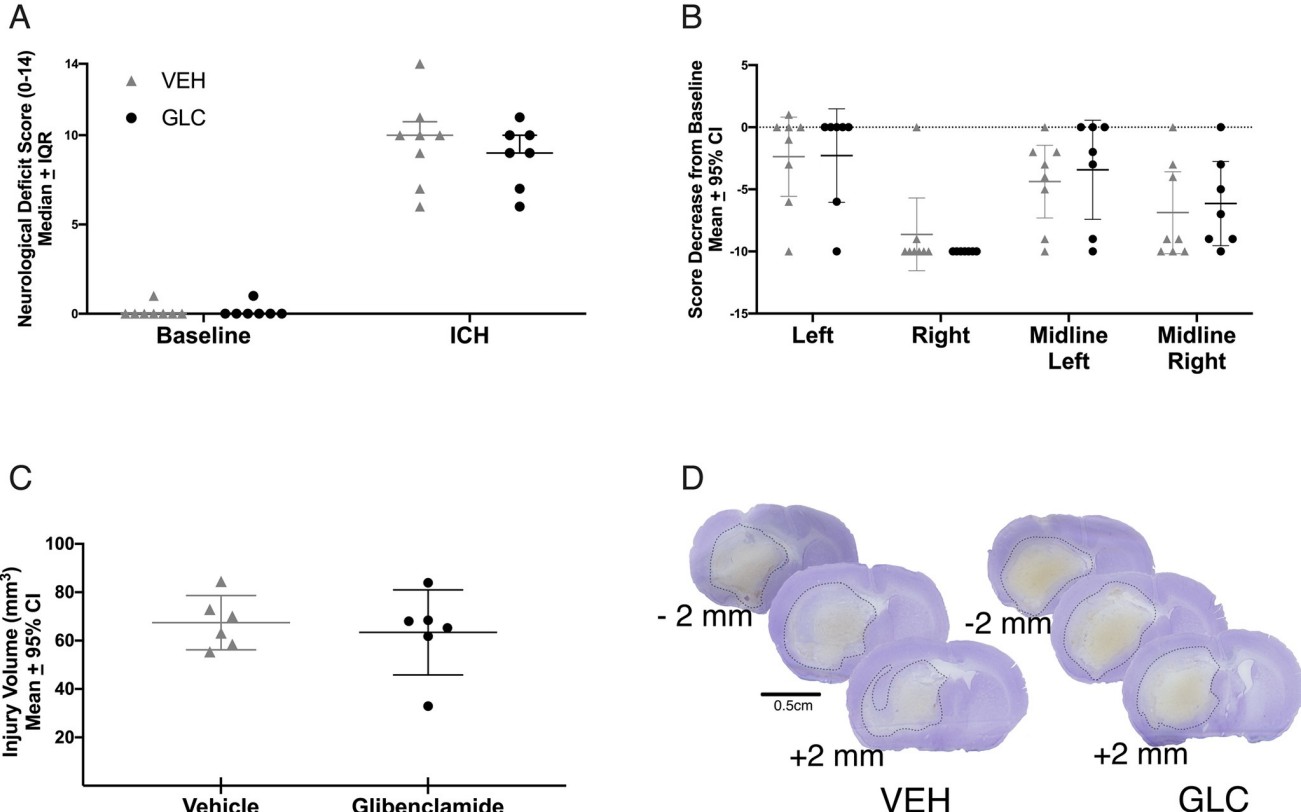

**Fig 6. Experiment 3 injury volume and behaviour.** No benefit of glibenclamide was found on either NDS (A, p = 0.5970), or forelimb placing of any paw (B, p≥0.4432). No difference was found in injury volume ± 4 mm around the maximum hematoma (C, p = 0.1280) between groups, reinforcing the results of experiment 1 that glibenclamide does not affect bleeding. Representative cresyl violet stained sections of -2 mm (anterior) to the maximum hematoma area, the maximum hematoma area, and +2 mm (posterior) to the maximum hematoma area are shown (D).

n = 26 VEH-S before exclusions). Since the only rats that received moderate strokes were in experiment 2, their behaviour scores were not re-analyzed, as they would be the same as reported in experiment 2.

**Body weight and mortality.**  To ensure no weight confounds were present between groups, animals were weighed before ICH (0 d) and at euthanasia (24 h) and pooled by severity. Weights were not significantly different between groups (p>0.1110), but rats with severe strokes experienced more weight loss than rats with moderate strokes (p = 0.0011) regardless of treatment.

Mortality was additionally pooled across experiments 1–4 (25% for VEH-S, 29.4% for GLC-S; no mortalities in moderate groups), broken down by severity, and no differences were found between severe stroke groups (p = 0.6783).

**Behaviour.**  Overall, ICH significantly worsened NDS after severe (p<0.0001 vs. baseline) stroke. Glibenclamide had no effect on NDS after pooling (Fig 8A, p = 0.7643). Severe ICH had no effect on left (ipsilateral) paw placement of VEH treated rats (Fig 8B, p = 0.0920). All other VEH placements (right, left midline, right midline) and all GLC paw placements performed worse after severe ICH (p≤0.0002). Glibenclamide did not affect forelimb placing of any paw after pooling (p≥0.1711). Next, all severe animals were ranked from 1 (best score; best outcome) to 62 (worst score) [45]. This was done for both NDS and

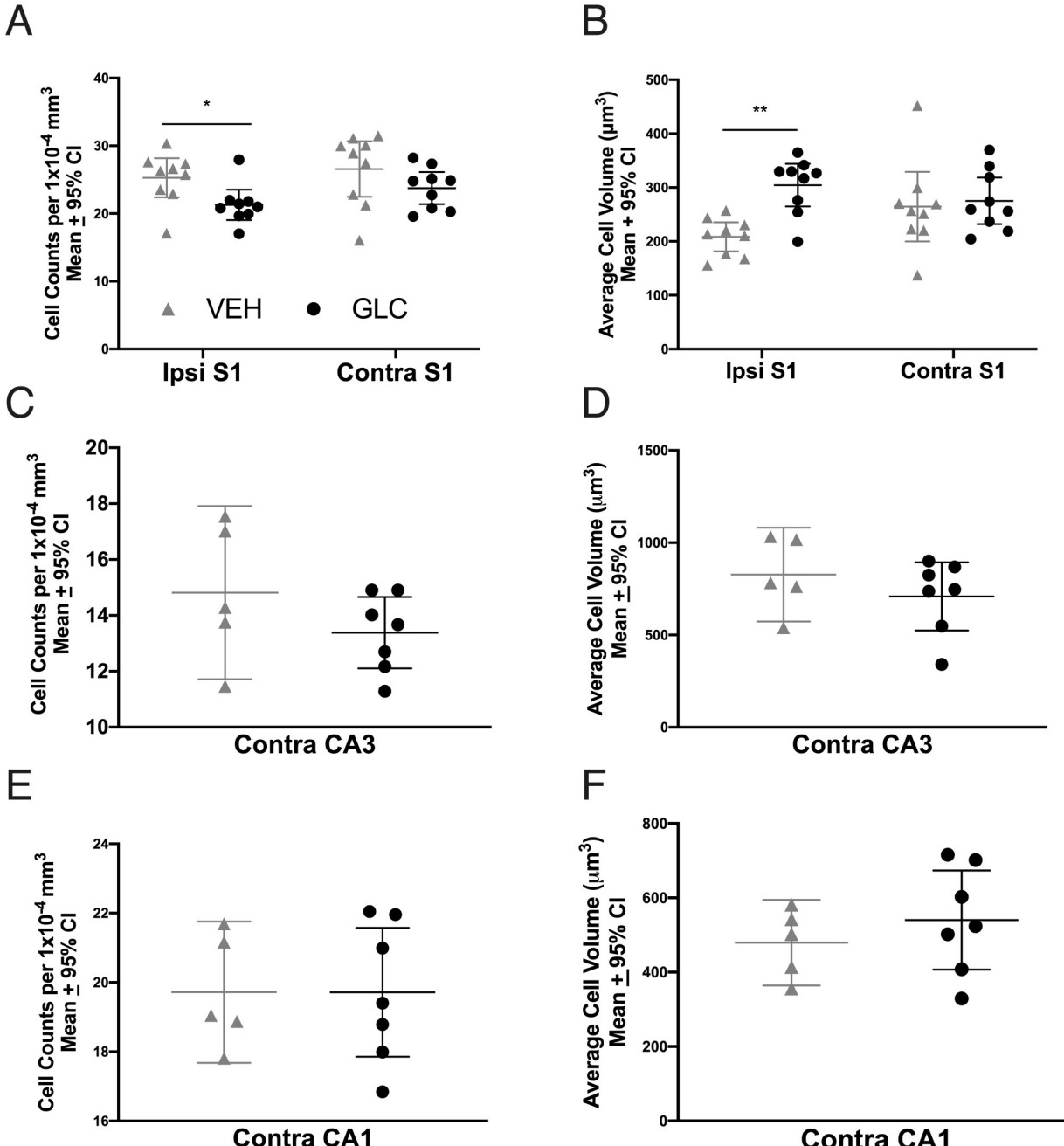

**Fig 7. Experiment 3 cell shrinkage.** In S1, glibenclamide had a large effect on cell density in the ipsilateral (Ipsi) injured hemisphere of glibenclamide treated rats, compared to the contralateral (Contra) hemisphere (A, p = 0.037). Additionally, glibenclamide had a large effect on cell volume in the ipsilateral hemisphere (B, p = 0.0017). No effect of glibenclamide was found on either cell density or volume in CA3 or CA1 (C-F, p≥0.2276), suggesting that this agent did not affect tissue compliance. Since brain water content was similar across groups, this indicates that glibenclamide worsened cytotoxic edema. * p<0.05, ** p<0.01.

forelimb placing of each paw except left, as ICH was not found to significantly affect paw placement (i.e., every rat received 4 ranks total). The scores were then averaged per rat. ICH caused significant impairment (Fig 8C, p = 0.0218) which was not affected by glibenclamide (p = 0.2180).

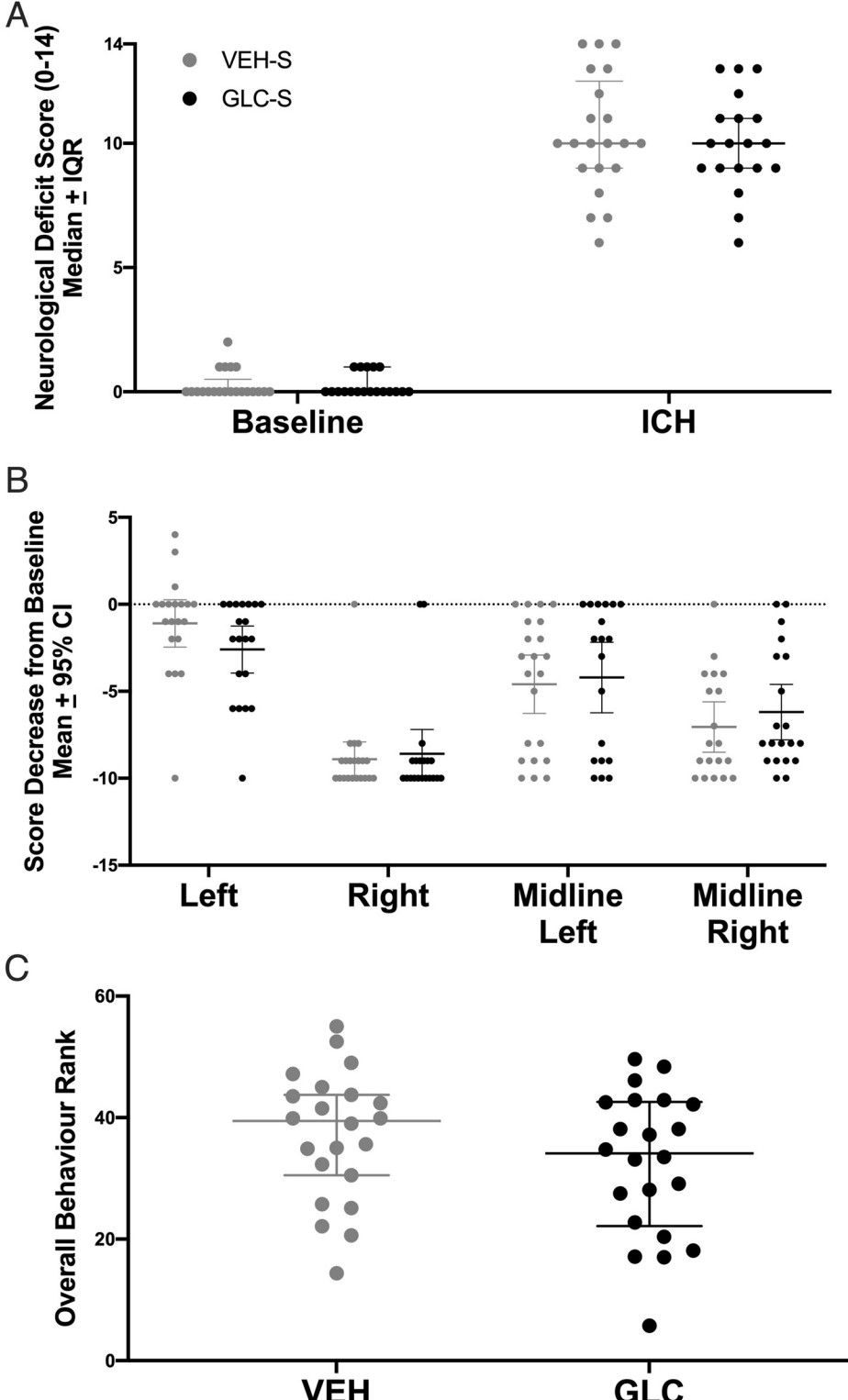

**Fig 8. Pooled behaviour.** After pooling, severe ICH worsened NDS performance (A, p<0.0001 vs. baseline), and all paw placements (B, p≤0.0002 vs. baseline) except for the left (ipsilateral) paw (p = 0.0920). For forelimb placing, a dashed line was set at 0, which indicates no change from baseline. After pooling, no meaningful benefit of glibenclamide was found on either NDS (p = 0.7643), forelimb placing of any paw (p≥0.1711), or overall behaviour rank (C, p = 0.2180). NDS and forelimb placing data for the moderate stroke group is in Fig 4.

### Experiment 4

**Mortality and exclusions.** For experiment 4, 2 animals died due to experimenter error (isoflurane overdose), and 3 animals (2 GLC, 1 VEH) experienced intra-ventricular hemorrhage rather than ICH; all were excluded from all analyses. Of the remaining, 9 animals spontaneously died, and 1 animal was prematurely euthanized for humane reasons (5 GLC, 5 VEH); all were excluded from all analyses except mortality. One animal was missing baseline data due to technical issues. To be conservative, his baseline was set as 1 (NDS) and 9 for each limb (forelimb placing) and was included in subsequent analyses. This inclusion did not change any conclusions.

**Behaviour.** ICH significantly worsened NDS after stroke (Fig 9A, p<0.0001 vs. baseline, Cohen's $d_z$ = 8.2 at 24 h, Cohen's $d_z$ = 45.2 at 72 h), and glibenclamide had no effect on NDS at either 24 h or 72 h (p>0.2063). ICH significantly worsened all forelimb placing at 24 h (Fig 9B, p≤0.0125 vs. baseline, Cohen's $d_z$ = 3.5). Deficits had resolved in the left forelimb (left and midline left, p≥0.107) by 72 h, but were still present in the right forelimb (right and midline right, p≤0.0415, Cohen's $d_z$ = 1.3). Glibenclamide had no effect on forelimb placing of any paw (p≥0.4589). In VEH rats, a significant improvement was found for midline left forelimb placing between 24 and 72 h (p = 0.0210), but not in GLC rats (p = 0.6384).

**Edema.** ICH significantly increased striatal brain water content to ~86% (vs. 79% contralaterally, p<0.0001, Cohen's $d_z$ = 10.9) and cortex brain water content to ~82% (vs. 79% contralaterally, p<0.0001, Cohen's $d_z$ = 29.5), but both were not affected by glibenclamide (Fig 9C, p≥0.4096).

### Microarray data

There was a significant effect of region (p = 0.030) but not group (p = 0.1573, interaction effect p = 0.4114) on *Abcc8* expression (codes for Sur-1) after untreated ICH, indicating differential expression of Sur 1 in cortex vs. hippocampus, but no effect of ICH on Sur1 expression. There was no effect of region (p = 0.4717), group (p = 0.1332), or interaction between region and group (p = 0.7338) on *Trpm4* expression after ICH (Fig 10). In total, our transcriptomics data discovered 2028 significantly up- or down-regulated genes in the peri-hematoma zone, and 483 in CA1, hence our methods were likely sensitive enough to detect major differences in channel expression.

## Discussion

Glibenclamide greatly reduces edema, behavioural deficits, and mortality in rodent models of severe ischemic stroke, and is currently in Phase III clinical trials [20, 23, 26–28]. Some have reported comparable benefits in rodent ICH models [26–28]. In this study, glibenclamide did not have a significant impact on any of our endpoints, despite using moderate and large bleeds that caused extensive edema. No difference in mortality or functional outcome was found, even after pooling across experiments. However, glibenclamide had no impact on hematoma volume or any of our safety measures (i.e., food and water intake, RGS, blood glucose), at least demonstrating that the drug is safe for administration as with current literature [29, 47].

In our past study, glibenclamide also had no effect on any of the endpoints [29]. This was hypothesized to be due to the small amount of edema present with a relatively modest bleed. For the present study, the average hematoma volume for the severe group was ~81 μL, much higher than previous glibenclamide studies in ICH, and ~3–6 × larger than our previous study [28, 29]. Based on an average brain weight of 2 g in rats and 1.4 kg in humans [48], this roughly translates to a severe ~57 mL bleed in humans. The comparable bleed for our moderate group would be a ~28 mL bleed in humans. Additionally, the 72-hour mortality rate for the severe

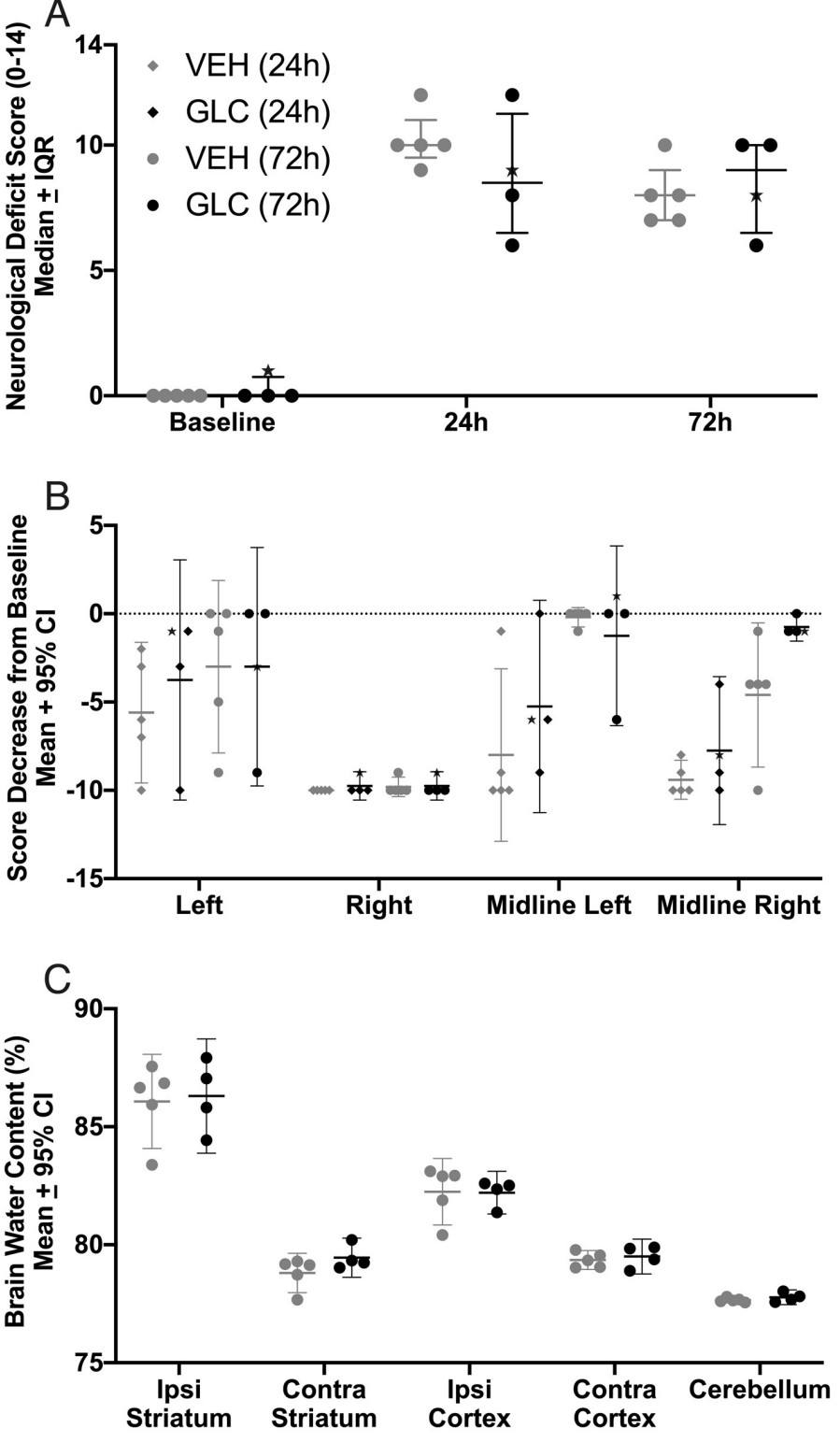

**Fig 9. Experiment 4 edema at 72 h.** ICH significantly worsened NDS performance (A, p<0.0001 vs. baseline, Cohen's $d_z$ = 8.2 at 24 h, Cohen's d = 45.2 at 72 h) and all forelimb placing at 24 h (B, p≤0.0125 vs. baseline, Cohen's $d_z$ = 3.5), for which the left forelimb deficits had resolved by 72 h (left and midline left, p≥0.107) but were still present in the right forelimb (right and midline right, p≤0.0415, Cohen's d = 1.3). For forelimb placing, the dashed line at 0 denotes no impairment. No meaningful benefit of glibenclamide was found on either NDS (p≥0.2063), or forelimb placing of

any paw (p≥0.4589). One rat had baseline data artificially set as 1 on the NDS and 9 for all forelimbs, and is indicated on all behaviour graphs by a star. Inclusion of this rat did not change our conclusions. ICH increased brain water content in both the ipsilateral striatum and cortex (C, p<0.0001), but was not affected by glibenclamide (p = 0.4096).

strokes was consistent with very severe strokes (55.6% for GLC, and 50% for VEH). Our moderate stroke group had no mortality, indicating the stroke was less severe, as expected. To further illustrate the severity of our stroke, histology showed all ICHs destroyed most of the ipsilateral hemisphere below the cortex, as is possible with very severe strokes. Lastly, in addition to severe strokes having more edema, the collagenase model also causes more edema than the autologous whole blood model [30–32]. Therefore, it is reasonable to assume that the severity of stroke and resultant edema was large enough to have given glibenclamide enough room to confer benefit.

Tissue compliance, or cell shrinkage as a result of raised ICP, is likely to be regulated by ion channels, perhaps including Sur1-Trpm4 [11, 12, 25]. Thus, we anticipated that glibenclamide may have a direct impact on cell volume throughout the brain after severe stroke. As well, we expected that glibenclamide could indirectly affect cell volume elsewhere in the brain by lowering edema in the peri-hematoma region and reducing ICP. In contrast to those expectations, unbiased stereology revealed that cell volume was significantly increased, while cell density was significantly reduced in ipsilateral S1 (peri-hematoma zone) in glibenclamide-treated rats. Since total percentage of brain-water content was similar between groups, this suggests that glibenclamide actually worsened cytotoxic edema in the peri-hematoma zone. Regardless, this effect did not notably affect behaviour or mortality, but it is concerning, as cytotoxic edema is known to contribute to greater vasogenic edema, higher ICP and worse injury [2, 14, 49]. We may not have been able to detect such effects simply because the extensive damage to white matter tracts and striatum effectively make it difficult to worsen behavioral outcome (i.e., floor effects). Further studies are needed to confirm the finding and to tease apart the mechanism. Notably, we have no simple explanation for why glibenclamide worsened cellular swelling in the ipsilateral hemisphere, especially since we did not detect evidence of Sur1-Trpm4 upregulation.

As noted, we did not observe differences in RNA expression of *Abcc8* or *Trpm4*, which code for the Sur1-Trpm4 channel, despite previous ischemic and hemorrhagic stroke studies reporting this [3, 24, 26, 28]. Since our transcriptomics analysis was able to detect significant differences in 2028 genes in the peri-hematoma zone, it is unlikely that our results were due to

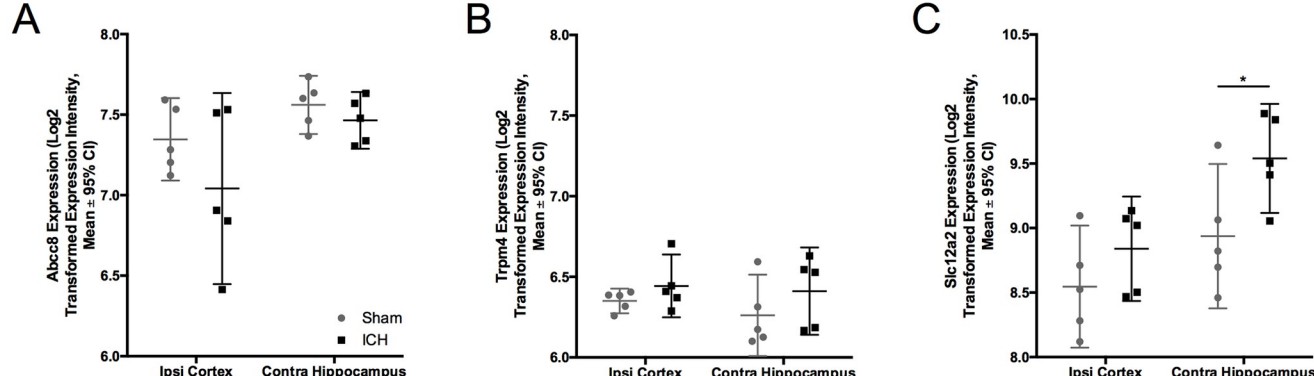

**Fig 10. Microarray data.** Rats were given ICH and mRNA expression of *Abcc8* (A), which codes for Sur1, and *Trpm4* (B) was investigated at a 24-hour survival time. ICH did not affect *Abcc8* (p = 0.1332) or *Trpm4* (p = 0.7338) expression.

insensitivity of our micro-array data. Although gene expression is not necessarily indicative of function, Sur1-Trpm4 is thought to be upregulated at 24 hours in the collagenase and autologous whole blood models of ICH [26, 28]. While the apparent lack of channel upregulation in our model of ICH can explain our lack of benefit with glibenclamide, it leaves unanswered why others find Sur1-Trpm4 channel upregulation while we did not. Possible reasons include species and strain differences, or different scoring methods. Zhou et al. and Jiang et al. found upregulation of Sur1-Trpm4 protein levels, perhaps indicating future studies should investigate channel protein expression more directly with Western Blot [26, 28]. Jiang et al. were additionally able to find upregulation of mRNA using RT-PCR rather than microarray [26]. This may indicate that other methods of assessing gene and protein expression may be needed to detect the upregulation of Sur1-Trpm4 than what was used in this study. As well, a few studies used a pre-treatment of glibenclamide, which we were not interested in due to our interest in using this agent acutely as a neuroprotective strategy [27, 28]. However, studies that found benefit used rats (strain unspecified) and the collagenase model [28], or C57/Bl6 mice and Sprague Dawley rats with the autologous whole blood model [26, 27]. Both Wilkinson et al. and this study used Sprague Dawley rats in the collagenase model [29]. Thus, there are no simple explanations for these differences among studies.

Unexpectedly, we found no significant difference in Gd concentration (BBB integrity) between the hemispheres, although there was a trend towards more Gd in the ipsilateral hemisphere. This may be related to the fact that increased BBB permeability occurs mostly in the peri-hematoma zone [50]. Due to the severity of the bleed, the examined striatal tissue was largely hematoma, which we would not expect Gd to enter in great quantities (i.e., the very low levels in the hematoma might counteract the higher levels in peri-hematoma tissue) [50]. Regardless, $Na^+$ and $K^+$ concentrations, which were affected by the stroke, were not significantly affected by glibenclamide administration. These ions are known to travel via the Sur1-Trpm4 channel and contribute to edema and BBB injury [51]. Thus, we had anticipated that blocking the Sur1-Trpm4 channel would alter $Na^+$ and $K^+$ concentrations, but this was not the case. Since measurements were done on bulk tissues, inferences cannot be made regarding intra- vs. extra-cellular ion concentrations, which may have been affected by treatment. One might also argue that a considerable portion of the edema occurring in our model resulted from serum extrusion, which the Sur1-Trpm4 channel should not affect. Nonetheless, there should still be considerable cytotoxic and vasogenic edema that glibenclamide could affect. Thus, the lack of effect with glibenclamide is likely in part explained by our findings that Sur1-Trpm4 channels were not upregulated following ICH, for unknown reasons. Future experiments should investigate this more directly with methods such as Western blot.

Our ability to test long-term benefit and outcome following stroke was limited by our short survival period, which was chosen to minimize suffering of animals who received the severe insult. However, our previous study did test long term effects and found no benefit, albeit with a modest-sized insult [46]. As well, glibenclamide is hypothesized to work primarily by reducing cytotoxic edema which has likely peaked by 24 hours in rats [2, 5–7], and other studies have found benefit of glibenclamide on edema with a 72 h endpoint.[26, 27]

The lack of treatment efficacy in this study should not be viewed as a failure to replicate (in the strictest sense) earlier positive findings as there are many differences among these studies (species, strain, etc.) [26–28]. Furthermore, and as with any negative drug study, some benefit or harm may have been missed, which might have been detected with another endpoint or a different measurement time. Nonetheless, our findings in this, and our previous study [29], strongly suggest that glibenclamide will not improve outcome in ICH. For this, we have used good scientific practice (blinding, randomization, power calculations, *a priori* planning, etc.), along with addressing multiple issues in translational research (use of several insult severities,

multiple endpoints, etc.) [52, 53]. Thus, these data should be considered in the broader context as a significant failure to find benefit in "confirmatory studies", and when sufficient data exist, they should be included in meta-analyses to help determine the true benefit of glibenclamide and whether there are certain contexts in which it provides biologically meaningful benefit [54]. The publishing of negative findings are critical to that enterprise [55].

## Supporting information

**S1 Dataset. All relevant data are within the manuscript and its S1 Dataset.**
(XLSX)

## Acknowledgments

The authors would like to thank Sherry Gu, Anna Kalisvaart, and Lane Liddle for their help and feedback on the manuscript, and Britt Fedor for her help with animal monitoring.

## Author Contributions

**Conceptualization:** Tiffany F. C. Kung, Cassandra M. Wilkinson, Glen C. Jickling, Frederick Colbourne.

**Data curation:** Tiffany F. C. Kung, Cassandra M. Wilkinson, Christine A. Dirks.

**Formal analysis:** Tiffany F. C. Kung, Cassandra M. Wilkinson, Glen C. Jickling.

**Funding acquisition:** Frederick Colbourne.

**Project administration:** Cassandra M. Wilkinson, Frederick Colbourne.

**Resources:** Frederick Colbourne.

**Supervision:** Cassandra M. Wilkinson, Glen C. Jickling, Frederick Colbourne.

**Writing – original draft:** Tiffany F. C. Kung, Christine A. Dirks.

**Writing – review & editing:** Tiffany F. C. Kung, Cassandra M. Wilkinson, Christine A. Dirks, Glen C. Jickling, Frederick Colbourne.

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
