## [Decision Letter · Decision Letter 0]

10 Feb 2021

PONE-D-20-33710

Glibenclamide does not improve outcome following severe collagenase-induced intracerebral hemorrhage in rats

PLOS ONE

Dear Dr. Colbourne,

Thank you for submitting your manuscript to PLOS ONE. After careful consideration, we feel that it has merit but does not fully meet PLOS ONE’s publication criteria as it currently stands. Therefore, we invite you to submit a revised version of the manuscript that addresses all of the points raised by expert reviewers below.

We look forward to receiving your revised manuscript.

Kind regards,

Vardan Karamyan, Pharm.D., Ph.D.

Academic Editor

PLOS ONE

2. As part of your revisions we kindly ask that you provide additional details about the care, use and welfare of the animals utilized for this study. Please update your Methods and Results section to address the following items: (1) all methods undertaken to minimize potential pain and distress, for instance: a description of your humane endpoints plan; (2) mortality rate during the study (if applicable); (3) unexpected adverse events; (4) monitoring parameters and so forth. Additionally, we ask that you please complete and submit the ARRIVE Guidelines 2.0 (Essential 10) checklist: https://arriveguidelines.org/resources/author-checklists.

Reviewers' comments:

Reviewer's Responses to Questions

**Comments to the Author**

1. Is the manuscript technically sound, and do the data support the conclusions?

Reviewer #1: Partly

Reviewer #2: Yes

2. Has the statistical analysis been performed appropriately and rigorously? 

Reviewer #1: Yes

Reviewer #2: Yes

3. Have the authors made all data underlying the findings in their manuscript fully available?

Reviewer #1: Yes

Reviewer #2: Yes

4. Is the manuscript presented in an intelligible fashion and written in standard English?

Reviewer #1: Yes

Reviewer #2: Yes

5. Review Comments to the Author

Reviewer #1: In the manuscript titled "Glibenclamide does not improve outcome following severe collagenase-induced intracerebral hemorrhage in rats," the authors report that glibenclamide has no beneficial effects after ICH. In some sections, the manuscript is not well edited, and the information is very limited because of the short survival time and sometimes small groups. In principle, non-efficacy of agents should also be published. However, the evidence of non-efficacy should also be done carefully and comprehensively. I am not sure if the statements made should be published in this league without further experiments.

1. in the introduction it is mentioned that especially in the ICH model used, brain edema is greatest 24-72h after the event. Nevertheless, 24h survival is chosen. The discussion also criticizes the survival time of the animals themselves. I would like to see at least 5 days survival from one arm of the experiment. With brain edema too low and stable BBB without gadopentetate dimeglumine differences, most likely from the severe ICH group.

2. The figures as scatterplot are to be welcomed. The figures are blurred and the labels are poorly legible. The group sizes with sometimes effectively only 5 animals per group clearly limit the significance of the results.

3. "Thus, the lack of effect with glibenclamide is best explained by our findings that Sur1-Trpm4 channels were not upregulated following ICH, for whatever reason." If you want to publish with more than 3 impact points, you should not leave a sentence as it is, but at least confirm this statement with another direct method (e.g. Western blot) or follow it up with in vitro methods if necessary.

4. Several minor errors, sources and material references in the methodology and the logical ductus of the discussion could be optimized.

Reviewer #2: • Looking at Fig 4A, there is a trend for better NDS for the Moderate ICH + Glc treated group. Please include a power analysis on this data to state what the sample sizes would need to be to detect a difference.

• The positive studies referenced for ICH and glibenclamide reported results at 72 hours. Since the dosing regimen of jiang et al is used, it is possible that glibenclamide is effective at 72 hours, but not 24. Xu et al., as stated, measured at 72 hrs. it is possible that glibenclamide takes longer than 22 hours to work. The work of zhou et al also at 72 showed less deficits. There are a few reasons why which need to be discussed. First, it may be that it takes longer than 24 hrs for glibenclamide to work. Second, the different findings may be due to the different scoring methods. Third, glibenclamide may need to be given pre-injury to have a significant benefit on the initial edema mechanisms. It is suggested to provide a 72 hr cohort to rule this out. Maybe also consider administering pre-ICH to test this.

6. PLOS authors have the option to publish the peer review history of their article (what does this mean?). If published, this will include your full peer review and any attached files.

Reviewer #1: No

Reviewer #2: No

---

## [Author Response · Author response to Decision Letter 0]

3 May 2021

Reviewer #1: In the manuscript titled "Glibenclamide does not improve outcome following severe collagenase-induced intracerebral hemorrhage in rats," the authors report that glibenclamide has no beneficial effects after ICH. In some sections, the manuscript is not well edited, and the information is very limited because of the short survival time and sometimes small groups.

• We have gone through the manuscript extensively and fixed any grammatical errors we could find. All sample sizes were decided on an a priori basis to have 80% power on our primary endpoints (lines 325 – 356) and our results were consistent across experiments and with our previous findings [1]. While one cannot use negative data to “prove” the null hypothesis, our findings do not support the alternative hypothesis. Furthermore, our data should be added to the growing literature in this area (e.g. for eventual meta-analysis) 

 In principle, non-efficacy of agents should also be published. However, the evidence of non-efficacy should also be done carefully and comprehensively. I am not sure if the statements made should be published in this league without further experiments.

1. in the introduction it is mentioned that especially in the ICH model used, brain edema is greatest 24-72h after the event. Nevertheless, 24h survival is chosen. The discussion also criticizes the survival time of the animals themselves. I would like to see at least 5 days survival from one arm of the experiment. With brain edema too low and stable BBB without gadopentetate dimeglumine differences, most likely from the severe ICH group.

• We initially chose our 24-hour endpoint as it is a time of cytotoxic edema, which glibenclamide is theorized to affect most and for ethical reasons to limit mortality as we would see with longer mortality times in our severe strokes. We recognize the need for a longer time-point and have conducted a study with a 72-hour endpoint to investigate edema, which we and others have shown to be a time of peak edema. This has been added to the manuscript as Experiment 4 (lines 557-596).

• We chose 72-hours rather than 5 days as edema peaks at this time-point, and previous studies that have found benefit of glibenclamide with edema have used 72-hour endpoints and this would allow the most direct comparison to other studies [1–3]. We found no benefits of glibenclamide on either striatal or cortical edema, supporting our results at 24 hours (lines 593-596). Further information on this study has been added to the manuscript as Experiment 4, and results are available for the reviewers to see in Figure 9 and the supplementary data file.

2. The figures as scatterplot are to be welcomed. The figures are blurred and the labels are poorly legible. The group sizes with sometimes effectively only 5 animals per group clearly limit the significance of the results.

• We believe this may be a problem with the PDF versions, as our original files are 600 DPI, the resolution cannot be further increased within the journal guidelines. Please download the images, as they should be of higher resolution. In our a priori power analyses were performed for all experiments to determine 80% power (lines 325 – 356). These group sizes were further increased to account for possible mortalities and exclusions. However, the severity of the stroke resulted in higher mortality than expected, therefore decreasing our group sizes (lines 527-533, 559-565). As adding more animals following loss due to mortality is bad practise, and our veterinarian had concerns with further experimentation at this level of injury [4], we reported our results as determined rather than adding more animals.

3. "Thus, the lack of effect with glibenclamide is best explained by our findings that Sur1-Trpm4 channels were not upregulated following ICH, for whatever reason." If you want to publish with more than 3 impact points, you should not leave a sentence as it is, but at least confirm this statement with another direct method (e.g. Western blot) or follow it up with in vitro methods if necessary.

• We agree with reviewer 1 and have expanded upon the original sentence (lines 700-710), and included a recommendation for future studies to investigate protein expression more directly than we were able to. Due to limitations imposed by COVID-19, it is not realistic for us to follow up with in vitro methods or Western blot at this time, but we have included this in our manuscript as a future direction (see lines 670-680, lines 709-710).

4. Several minor errors, sources and material references in the methodology and the logical ductus of the discussion could be optimized.

• We have re-reviewed the manuscript in greater detail and have edited it throughout. We have condensed references where possible throughout the manuscript. All references in the manuscript have been checked, and the discussion has been changed following inclusion of Experiment 4.

 

Reviewer #2: • Looking at Fig 4A, there is a trend for better NDS for the Moderate ICH + Glc treated group. Please include a power analysis on this data to state what the sample sizes would need to be to detect a difference.

• The p-value for this comparison was p=0.056 as determined by a Mann-Whitney analysis. With an effect size of 0.97 from our study and an alpha of 0.05, group sizes of 19 per group would be needed to obtain 0.811 power. Alternatively, with our groups of n=9 and no mortalities or exclusions, we had a post-hoc power of 0.471. Since there were no differences in forelimb placing for the moderate group, or for behaviour at all in the general group, this effect may not be biologically meaningful. All this information has been added to the manuscript under the section for Experiment 2, Behaviour (lines 426-433). 

• The positive studies referenced for ICH and glibenclamide reported results at 72 hours. Since the dosing regimen of jiang et al is used, it is possible that glibenclamide is effective at 72 hours, but not 24. Xu et al., as stated, measured at 72 hrs. it is possible that glibenclamide takes longer than 22 hours to work. The work of Zhou et al also at 72 showed less deficits. There are a few reasons why which need to be discussed. First, it may be that it takes longer than 24 hrs for glibenclamide to work. 

• While it’s true that a bulk of the literature is focussed on 72 hours, there have been a number of studies that have shown effects/benefit from glibenclamide at 24 hours. Firstly, Jiang et al., 2017 [2] and Zhou et al., 2018 [5] both found Sur1 upregulation at 24-hours, suggesting glibenclamide should be able to act as soon as 24-hours post stroke. Jiang et al., 2017 and Sheth et al., 2016 (in the GAMES-RP trial) both found that glibenclamide reduced MMP-9 expression at 24 hours, showing glibenclamide acts within 24 hours post-stroke [2,6]. Lastly, Zhou et al., 2018 found a significantly decreased mNSS 24-hours post-stroke in glibenclamide-treated animals [5]. Studies done on glibenclamide following are comparatively scarce compared to ischemic stroke, but of the few studies done there has been evidence that glibenclamide should be able to act and potentially confer benefit within 24 hours from stroke.

• However, following the reviewers comments we have understood the importance of looking at a 72-hour endpoint. Thus, we conducted a study investigating edema at 72-hours, as reviewer 2 suggested, as this is the peak of edema, and when other studies found benefit of the drug [2,3]. We did not find any benefit of glibenclamide on either striatal or cortical edema. These findings have been added to the manuscript as Experiment 4 (lines 557-596), and are available in figure 9 as well as the supplementary file dataset.

Second, the different findings may be due to the different scoring methods. 

• We agree with reviewer 2 and this point has been added to the limitations section (line 671).

Third, glibenclamide may need to be given pre-injury to have a significant benefit on the initial edema mechanisms. It is suggested to provide a 72 hr cohort to rule this out. Maybe also consider administering pre-ICH to test this.

• The purpose of this study was to investigate glibenclamide as a post-stroke treatment to be administered to patients in an attempt at neuroprotection. Since damage in the collagenase model occurs over hours following the injection, it is likely that our 2h post-ICH administration is being given prior to the full extent of damage. Further, a pre-treatment group would be restricted to diabetic patients on the drug prior to stroke, and we were not interested in this aspect due to our interest in using this agent acutely as a neuroprotective agent (see lines 108-109, 680-684). We agree with reviewer 2 that pre-treating with glibenclamide may result in different outcomes, and this point has been added to the limitations section (lines 680-684).

 

1. Wilkinson CM, Brar PS, Balay CJ, Colbourne F. Glibenclamide, a Sur1-Trpm4 antagonist, does not improve outcome after collagenase-induced intracerebral hemorrhage. Arai K, editor. PLoS ONE. 2019;14: e0215952. doi:10.1371/journal.pone.0215952

2. Jiang B, Li L, Chen Q, Tao Y, Yang L, Zhang B, et al. Role of Glibenclamide in Brain Injury After Intracerebral Hemorrhage. Transl Stroke Res. 2017;8: 183–193. doi:10.1007/s12975-016-0506-2

3. Xu F, Shen G, Su Z, He Z, Yuan L. Glibenclamide ameliorates the disrupted blood–brain barrier in experimental intracerebral hemorrhage by inhibiting the activation of NLRP3 inflammasome. Brain Behav. 2019;9: e01254. doi:10.1002/brb3.1254

4. Wicherts JM, Veldkamp CLS, Augusteijn HEM, Bakker M, van Aert RCM, van Assen MALM. Degrees of Freedom in Planning, Running, Analyzing, and Reporting Psychological Studies: A Checklist to Avoid p-Hacking. Front Psychol. 2016;7. doi:10.3389/fpsyg.2016.01832

5. Zhou F, Liu Y, Yang B, Hu Z. Neuroprotective potential of glibenclamide is mediated by antioxidant and anti-apoptotic pathways in intracerebral hemorrhage. Brain Research Bulletin. 2018;142: 18–24. doi:10.1016/j.brainresbull.2018.06.006

6. Sheth KN, Elm JJ, Beslow LA, Sze GK, Kimberly WT. Glyburide Advantage in Malignant Edema and Stroke (GAMES-RP) Trial: Rationale and Design. Neurocrit Care. 2016;24: 132–139. doi:10.1007/s12028-015-0189-7

---

## [Editor Report · Decision Letter 1]

19 May 2021

Glibenclamide does not improve outcome following severe collagenase-induced intracerebral hemorrhage in rats

PONE-D-20-33710R1

Dear Dr. Colbourne,

We’re pleased to inform you that your manuscript has been judged scientifically suitable for publication and will be formally accepted for publication once it meets all outstanding technical requirements.

Kind regards,

Vardan Karamyan, Pharm.D., Ph.D.

Academic Editor

PLOS ONE
---

## [Editor Report · Acceptance letter]

24 May 2021

PONE-D-20-33710R1 

Glibenclamide does not improve outcome following severe collagenase-induced intracerebral hemorrhage in rats 

Dear Dr. Colbourne:

I'm pleased to inform you that your manuscript has been deemed suitable for publication in PLOS ONE. Congratulations! Your manuscript is now with our production department. 

Kind regards, 

on behalf of

Dr. Vardan Karamyan 

Academic Editor

PLOS ONE